# Single-cell analysis reveals that cryptic prophage protease LfgB protects *Escherichia coli* during oxidative stress by cleaving antitoxin MqsA

Laura Fernández-García,[1,2,3] Xinyu Gao,[4,5,6,7] Joy Kirigo,[1] Sooyeon Song,[1,8,9] Michael E. Battisti,[1] Rodolfo Garcia-Contreras,[10] Maria Tomas,[2,3] Yunxue Guo,[4,5,6,7,11] Xiaoxue Wang,[4,7,11] Thomas K. Wood[1]

ABSTRACT   Although toxin/antitoxin (TA) systems are ubiquitous, beyond phage inhibition and mobile element stabilization, their role in host metabolism is obscure. One of the best-characterized TA systems is MqsR/MqsA of *Escherichia coli*, which has been linked previously to protecting gastrointestinal species during the stress it encounters from the bile salt deoxycholate as it colonizes humans. However, some recent whole-population studies have challenged the role of toxins such as MqsR in bacterial physiology since the *mqsRA* locus is induced over a hundred-fold during stress, but a phenotype was not found upon its deletion. Here, we investigate further the role of MqsR/MqsA by utilizing single cells and demonstrate that upon oxidative stress, the TA system MqsR/MqsA has a heterogeneous effect on the transcriptome of single cells. Furthermore, we discovered that MqsR activation leads to induction of the poorly characterized *yfjXY ypjJ yfjZF* operon of cryptic prophage CP4-57. Moreover, deletion of *yfjY* makes the cells sensitive to $H_2O_2$, acid, and heat stress, and this phenotype was complemented. Hence, we recommend *yfjY* be renamed to *lfgB* (less fatality gene B). Critically, MqsA represses *lfgB* by binding the operon promoter, and LfgB is a protease that degrades MqsA to derepress *rpoS* and facilitate the stress response. Therefore, the MqsR/MqsA TA system facilitates the stress response through cryptic phage protease LfgB.

IMPORTANCE   The roles of toxin/antitoxin systems in cell physiology are few and include phage inhibition and stabilization of genetic elements; yet, to date, there are no single-transcriptome studies for toxin/antitoxin systems and few insights for prokaryotes from this novel technique. Therefore, our results with this technique are important since we discover and characterize a cryptic prophage protease that is regulated by the MqsR/MqsA toxin/antitoxin system in order to regulate the host response to oxidative stress.

KEYWORDS   toxin/antitoxin systems, cryptic prophage, stress response

Toxin/antitoxin (TA) systems are encoded in the genomes of nearly all archaea and bacteria and are classified into eight main types based on how the antitoxin inactivates the toxin (1). We discovered that phage inhibition is one of the primary physiological roles of TA systems and determined that the mechanism is toxin induction via host transcription shutdown by the attacking phage (2); these results were confirmed 25 years later (3). TA systems also stabilize mobile genetic elements (4–7). Beyond these two functions, there is controversy regarding the physiological roles of TA systems (8).

The MqsR/MqsA TA system was discovered as induced in a biofilm transcriptome study (9) and shown to be a TA system using the structures of the toxin (MqsR) and antitoxin (MqsA) (10). MqsR degrades mRNA with the 5′-GCU site (11), and MqsA was

Editor Ilana Kolodkin-Gal, The Hebrew University of Jerusalem, Rehovot, Israel

Ad Hoc Peer Reviewer Robert J. C. McLean, Texas State University, San Marcos, Texas, USA

Address correspondence to Thomas K. Wood, twood@engr.psu.edu.

The authors declare no conflict of interest.

See the funding table on p. 11.

[This article was published on 11 January 2024 with the incorrect supplemental file. The supplemental material was corrected in the current version, posted on 12 January 2024.]

found not only to regulate its own promoter but also to repress the oxidative stress response via DNA binding at a palindrome upstream of the stress response sigma factor RpoS (12) and to repress curli synthesis by binding to the promoter of the gene that encodes the master biofilm regulator CsgD (13). Moreover, MqsR/MqsA controls the TA system GhoT/GhoS as a cascade (14) and helps *Escherichia coli* colonize the gastrointestinal (GI) tract by surviving bile acid stress (15); activation of toxin MqsR during bile stress leads to degradation of YgiS mRNA, and this transcript encodes a periplasmic protein that promotes bile uptake. Furthermore, several groups have linked MqsR/MqsA to antibiotic tolerance based upon deletion of *mqsR* (16–18), and MqsR/MqsA has been linked to heat shock (19), biofilm formation (20), nitrogen starvation (21), and nitric oxide (22) in *E. coli* and copper stress (23), vesicles (24), and biofilm formation (25) in *Xylella fastidiosa* as well as biofilm formation in *Pseudomonas fluorescens* (26) and persistence and biofilm formation in *Pseudomonas putida* (27).

In contrast to these myriad results with MqsR/MqsA, a report based on negative results claimed that the *E. coli* MqsR/MqsA TA system has no role in stress resistance, based on a lack of induction of the *mqsRA* locus and a lack of phenotype upon deleting *mqsRA* (28). Strikingly, these transcription results were invalidated within a few months as *mqsRA* transcription in the wild-type strain was shown to increase by over 181-fold during amino acid stress and 90-fold during oxidative stress (29). This work (29) also claimed that there was no physiological effect of deleting *mqsRA*, but, unfortunately, they utilized a TA deletion strain that has substantial non-related mutations, including large chromosomal inversions (30); utilization of TA deletion strains with many coding errors beyond those of the TA systems has led to notorious retractions in the TA field, as we have summarized previously (31). Critically, their claim (29) of a lack of a physiological role of MqsR/MqsA was undercut by their later results which showed MqsR/MqsA/MqsC inhibited T2 phage (32). We have confirmed these results and shown that phage inhibition by MqsRAC induces persistence rather than abortive infection (33). Furthermore, these groups (28, 29) used strains with "both" MqsR and MqsA inactivated rather than studying the effect of either the toxin or antitoxin alone, that is, MqsR and MqsA work "together" during the oxidative stress response.

Based on these inconsistencies, we hypothesized that a better approach, due to heterogeneous gene expression (34), would be to investigate the impact of MqsR/MqsA on cell physiology by monitoring the transcriptome of "single cells" since all previous studies have been based on population averages. Single-cell transcriptomic studies have been initiated by several labs (34–38), and here, we utilized the high-throughput microfluidic approach that relies on labeling each transcript with unique 50-nt single-stranded DNA probes to determine the impact of inactivation of MqsR/MqsA during oxidative stress (38). We chose oxidative stress as the representative insult to cells since both anaerobes and aerobes must deal with this nearly universal stress (39), and MqsA has been shown to negatively regulate the oxidative stress response (12). Using this approach, we determined that the *lfgABCDE* operon (formerly the uncharacterized operon *yfjXY ypjJ yfjZF*) of cryptic prophage CP4-57 is induced in single cells and that LfgB is a protease that is repressed by antitoxin MqsA and degrades MqsA to activate the *E. coli* stress response through sigma factor RpoS.

## RESULTS

### Antitoxin MqsA reduces the population stress response

We first investigated whether deleting an unmarked *mqsRA* mutation affected the response of *E. coli* to oxidative stress (20 mM $H_2O_2$ for 10 min) and found that, for the whole population, the wild-type cells were *more* sensitive to $H_2O_2$ (85% ± 15% death for the wild type vs 55% ± 10% for *mqsRA*). Similar population-wide results were seen with acid stress (pH 2.5 for 10 min for four cycles), where the wild-type strain was 64 times more sensitive. These results agree well with our previous results showing that antitoxin MqsA represses *rpoS* by binding at a palindrome to help regulate stress resistance (12). We note that the $H_2O_2$ and acid phenotypes of a *mqsRA* mutant were

complemented previously and production of MqsA reduces peroxidase activity (12). Moreover, our results suggest that inactivating toxin MqsR should reduce viability by elevating MqsA concentrations since the additional antitoxin MqsA will repress *rpoS*, and as expected, when *mqsR* is deleted, cells are 14 ± 6 times more sensitive than the wild type to oxidative stress. Therefore, the *mqsRA* mutant is better prepared to withstand oxidative and acid stresses as its stress response via RpoS is activated due to the absence of the repressor MqsA.

## Single-cell analysis reveals that LfgB increases cell viability during oxidative stress

Using single cells, we further investigated the role of MqsR/MqsA during oxidative stress by comparing the wild-type strain vs the unmarked *mqsRA* mutant in single cells. Utilizing 20 mM $H_2O_2$ for 10 min, we found (Table 1) that several cryptic prophage genes are induced in the wild-type strain relative to the *mqsRA* mutant, including *lfgA* of the *lfgABCDE* operon; previously, LfgD (YfjZ) of this operon was shown by us to enhance MqsR toxicity (40). Furthermore, the induction of two genes that encode heat-shock proteins (*ibpAB*) and one gene that encodes an osmotic stress response protein (*yciF*) served as positive controls for our single-cell analysis. Note that these results required the single-cell approach as changes in the *lfg* operon were not detected using population averages (Table S1A).

Based on the single-cell transcriptome results, we tested 10 knockouts of the most highly induced genes and found that the *lfgA* deletion nearly completely prevented cells from surviving 20 mM $H_2O_2$ for 10 min (99.990% ± 0.004% death), whereas the wild-type strain had only 14% ± 13% death. Since we were unable to complement this phenotype by producing LfgA in a *lfgA* deletion mutant, we investigated whether a polar mutation was involved via kanamycin insertion into *lfgA* by investigating the next gene downstream of *lfgA, lfgB* (Fig. S1; Table 2), and found that deletion of *lfgB* also prevents survival with 20 mM $H_2O_2$ for 10 min (91% ± 8% death); this phenotype could be complemented by producing LfgB from pCA24N-*lfgB* (Table 2). Moreover, since RpoS positively controls the KatG/KatE catalase activity (12), these results were confirmed by observing the oxygen bubbles produced from catalase activity after incubating the *lfgB* mutant and complemented strain for 10 min with 20 mM of $H_2O_2$ (Fig. S2); quantifying the catalase results, the *lfgB* mutation reduced catalase activity by 60% ± 40%, and producing LfgB from pCA24N-*lfgB* nearly completely restored the catalase activity (95% ± 5%, Fig. 1A). Hence, we focused on LfgB to determine its role with MqsR/MqsA.

Since RpoS also controls the heat (41) and acid response (42) in *E. coli*, we hypothesized that inactivating LfgB should reduce viability after heat and acid treatments. Consistent with the reduction in the oxidative stress response, we found that the *lfgB* deletion reduces survival during acid (pH 4.0 for 10 min) stress (39% ± 5% death for *lfgB* vs 28% ± 4% death for wild type), as well as during heat (30 min at 50°C) stress (18% ± 12% death for *lfgB* vs 0% ± 7% death for wild type). Both phenotypes were complemented by producing LfgB from pCA24N-*lfgB* (Table 1).

Note that LfgB does not play a role in persister cell formation since, for survival after 3 h with ampicillin at 10× the minimum inhibitory concentration, there was little difference in cell viability (0.8% ± 0.4% viable for wild type vs 0.4% ± 0.2% viable for *lfgB*). Hence, LfgB is important for the stress response rather than antibiotic persistence. Also, deleting *lfgB* reduces the growth rate in lysogeny broth (LB) medium by 25% (1.2 ± 0.1/h vs 1.6 ± 0.2/h), so the dramatic reduction in viability of the *lfgB* mutant in the presence of $H_2O_2$ is not a result of poor growth.

LfgB is a poorly characterized protein of cryptic prophage CP4-57 whose production previously led to a mutator phenotype (43). To understand the relationship of this protein with the MqsR/MqsA TA system, we analyzed the RNA structure of the operon, finding two possible 5′-GCU sites accessible to toxin MqsR for *lfgB* (44) in the predicted minimum free energy (MFE) structure for whole operon mRNA (Fig. S3A), which are not

**TABLE 1** Impact on gene expression after inactivating the MqsR/MqsA TA system in *E. coli* during oxidative stress[a]

| Gene | Cluster | WT | ΔmqsRAΔkan | Gene | Cluster | WT | ΔmqsRAΔkan |
|------|---------|------|------------|------|---------|-------|------------|
| *yneL* | 1 | −0.7 | −0.9 | *yciF* | 1 | −0.26 | −0.83 |
| | 2 | −4.4 | 1.5 | | 2 | −5.12 | 2.15 |
| | 3 | 2.3 | 1.9 | | 3 | **8.76** | 0.23 |
| | 4 | 2.93 | 0.28 | | 4 | 3.39 | 0.39 |
| | 5 | 3.08 | 0.43 | | 5 | 3.54 | 0.54 |
| | 6 | **9.21** | 0.53 | | 6 | 3.89 | 0.64 |
| | 7 | 3.54 | | | 7 | 4 | |
| *gatR* | 1 | −0.07 | 1.82 | *yoeA* | 1 | 1.15 | −1.57 |
| | 2 | −4.93 | −0.51 | | 2 | −4.93 | 2.04 |
| | 3 | 2.95 | 0.93 | | 3 | 2.95 | 1.12 |
| | 4 | **9.2** | 1.09 | | 4 | 3.59 | 1.28 |
| | 5 | 3.73 | 1.24 | | 5 | 3.73 | 1.43 |
| | 6 | 4.08 | 1.34 | | 6 | 4.08 | 2.75 |
| | 7 | 4.2 | | | 7 | **8.59** | |
| *lfgA* | 1 | 0.74 | −0.57 | *ydiL* | 1 | 0.74 | 0.38 |
| | 2 | −4.12 | 1.88 | | 2 | −4.12 | 0.56 |
| | 3 | 3.76 | 1.93 | | 3 | 3.76 | 0.02 |
| | 4 | 4.39 | 2.09 | | 4 | 4.39 | 0.19 |
| | 5 | 4.54 | 2.24 | | 5 | **8.54** | 0.33 |
| | 6 | 4.89 | 2.34 | | 6 | 4.89 | 1.53 |
| | 7 | **9** | | | 7 | 5 | |
| *yagA* | 1 | 0.15 | −0.72 | *ibpA* | 1 | 0.162 | −0.068 |
| | 2 | −4.71 | 1.3 | | 2 | −0.166 | −0.404 |
| | 3 | 3.18 | 1.12 | | 3 | 0.008 | 1.272 |
| | 4 | **8.98** | 1.28 | | 4 | 0.16 | 1.218 |
| | 5 | 4.22 | 2.65 | | 5 | 0.196 | 0.106 |
| | 6 | 4.57 | 1.53 | | 6 | 0.11 | −0.334 |
| | 7 | 4.68 | | | 7 | −0.092 | |
| *holE* | 1 | −0.43 | −2.31 | *ibpB* | 1 | 0.632 | −0.55 |
| | 2 | −4.12 | 3.62 | | 2 | −0.666 | 1.034 |
| | 3 | 2.59 | 1.6 | | 3 | 0.398 | 0.974 |
| | 4 | 3.22 | 1.77 | | 4 | 0.574 | 0.862 |
| | 5 | 3.37 | 1.92 | | 5 | 0.466 | 0.898 |
| | 6 | **8.89** | 2.02 | | 6 | 0.726 | 0.896 |
| | 7 | 3.83 | | | 7 | 0.322 | |

[a]Genes with the highest and lowest expressions in the single-cell transcriptomic analysis are indicated after treating exponentially growing cells with 20 mM $H_2O_2$ for 10 min. WT is BW25113. Largest values indicated by bold text.

available in the MFE-predicted structure of only the transcript containing just *lfgB* mRNA (Fig. S3B). Hence, MqsR may degrade the mRNA containing *lfgB*.

## MqsA binds the *lfgA* promoter

We also considered the possibility that MqsA regulates the *lfg* operon by binding at its palindromic sequence 5′-ACCT N (2, 6) AGGT upstream of the promoter as shown previously for the *mqsRA*, *csgD*, and *rpoS* promoters (12, 13, 45). We found a probable MqsA palindromic sequence, 5′-ACCG (N5) CGGT, (gray highlight) 162 bp upstream of the start codon of *lfgA* (Fig. S4). Thus, we hypothesized that MqsA represses transcription of the operon and overproduced MqsA from pCA24N-*mqsA* and observed that *lfgA* and *lfgB* are repressed 4 ± 1- and 3 ± 0.6-fold, respectively (Table S1B). However, using electrophoretic mobility shift assay (EMSA), we found that mutating the *lfgA* promoter to interrupt this MqsA palindrome did not affect MqsA binding (Fig. 1B). Hence, we conducted a

**TABLE 2** Phenotypes of BW25113 (WT) and BW25113 Δ*lfgB* under different stresses[a]

| Condition | Strain | % death | SD | Ratio |
|---|---|---|---|---|
| $H_2O_2$ | WT | 14 | 10 | 1 |
| | Δ*lfgB* | 91 | 8 | 6.4 |
| | Δ*lfgB*/pCA24N | 64 | 20 | 4.6 |
| | Δ*lfgB*/pCA24N-*lfgB* | 26 | 20 | 1.8 |
| Acid | WT | 28 | 4 | 1 |
| | Δ*lfgB* | 39 | 5 | 1.4 |
| | Δ*lfgB*/pCA24N | 35 | 4 | 1.3 |
| | Δ*lfgB*/pCA24N-*lfgB* | 25 | 2 | 0.9 |
| Heat | WT | −13 | 7 | 1 |
| | Δ*lfgB* | 18 | 11 | −1.4 |
| | Δ*lfgB*/pCA24N | 21 | 3 | −1.6 |
| | Δ*lfgB*/pCA24N-*lfgB* | −14 | 3 | 1.0 |

[a]SD indicates standard deviation

DNA-footprinting assay (Fig. S5) and determined that the MqsA-binding site is 245 bp upstream of the start codon, with a putative palindromic sequence 5′-ACAT (N2) ACAT (green highlight) (Fig. S4). Inactivating this MqsA-binding site in the *lfgA* promoter region via mutation confirmed the DNA-footprinting results since MqsA binding was abolished as shown by EMSA (Fig. 1B). Hence, MqsA, a known regulator, binds the promoter of the operon containing *lfgB*.

## LfgB controls the $H_2O_2$ response likely through MqsA degradation

To gain further insights into how LfgB interacts with the MqsR/MqsA TA system, we analyzed the protein structure of *lfgB*. Critically, LfgB is a putative zinc protease based on its predicted structure (UniProtKB: P52140), with a Mpr1, Pad1 N-terminal domain (residues 38–160) and a JAB1/MPN/Mov34 metalloenzyme motif (metalloprotease-like zinc site) (46, 47) (Fig. 1C). Based on this predicted structure, we purified LfgB and tested its protease activity against purified MqsA and found that LfgB degrades MqsA after overnight incubation at 37°C (Fig. 1D). In addition, we found that LfgB shows protease activity on α-casein using Lon protease as a positive control (Fig. S6), although we cannot strictly rule out the other proteases present in the purified LgfB purification. Unfortunately, the solubility of LfgB is extremely low, and we were unable to improve its solubility after many attempts, including purification under denaturing conditions and fusing small ubiquitin-like modifier and glutathione S-transferase tags to LfgB. However, mass spectroscopy clearly shows that LfgB was purified successfully, which decreases the likelihood of background protease activity, and indicates that LfgB likely digests itself to a significant degree (Fig. S7). These results also suggest that LfgB may be a membrane protein.

Further proof of MqsA degradation by LfgB was shown by the threefold induction of *rpoS* when LfgB is produced during $H_2O_2$ stress (Table S1C). This induction is likely due to the degradation of MqsA by LfgB, which allows for the production of RpoS since MqsA represses the *rpoS* promoter by binding at a conserved palindrome (12). Corroborating this, inactivating RpoS reduced the impact of LfgB on catalase activity (Fig. 1A).

## DISCUSSION

Here, using the single-cell transcriptome for the first time to study TA systems, we determined additional insights into how the MqsR/MqsA Type II TA system is physiologically important for the growth of *E. coli* during exposure to $H_2O_2$ stress. Specifically, we (i) identified that the *lfg* operon of cryptic prophage CP4-57 is induced during oxidative stress in single cells, (ii) found that MqsA represses the *lfg* operon, and (iii) characterized LfgB as a protease that degrades antitoxin MqsA. Remarkably, our results demonstrate that the cell combines the tools of its former enemy, prophage CP4-57, with those of the MqsR/MqsA TA system, to regulate its stress response.

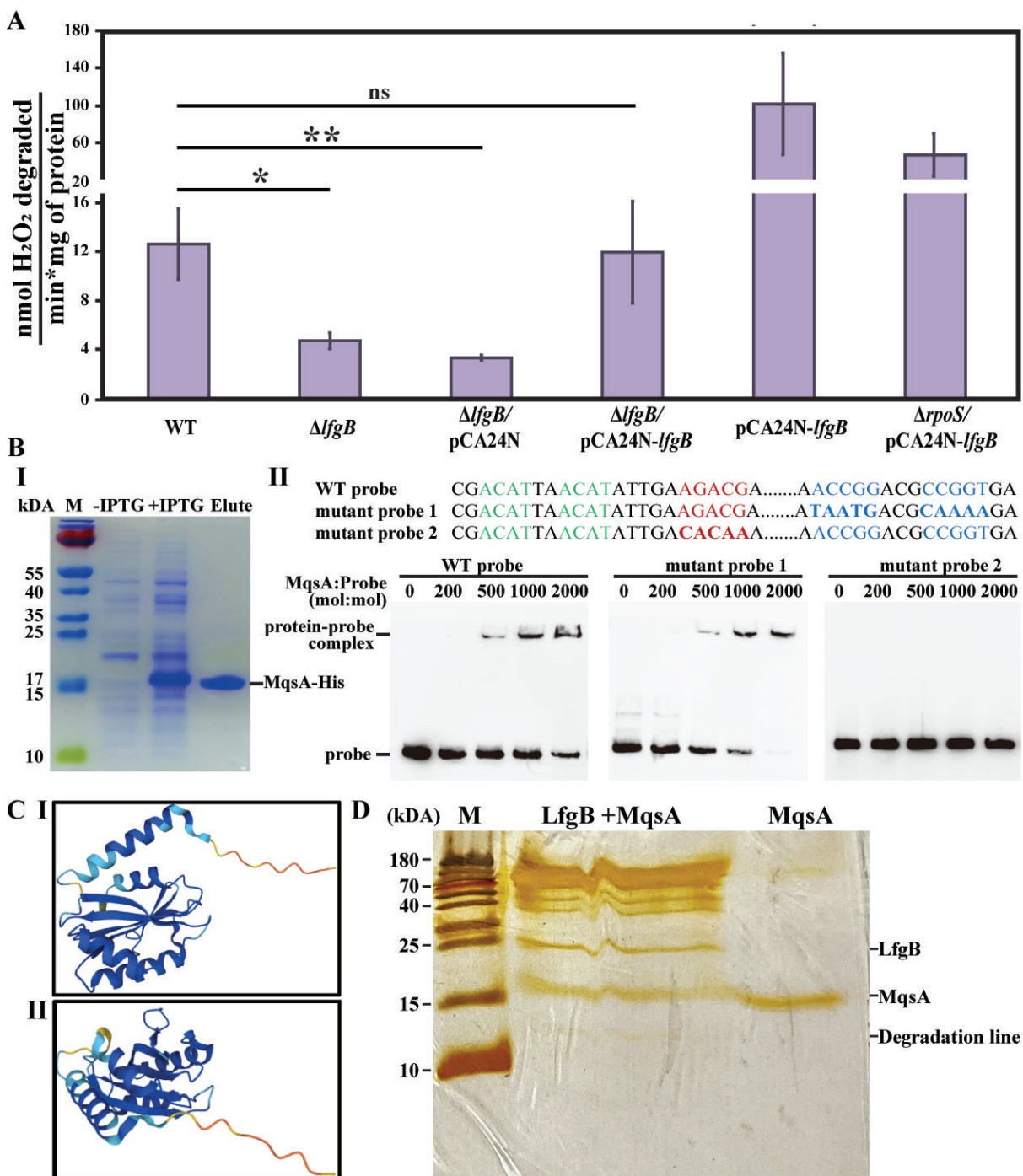

**FIG 1** (A) LfgB increases catalase activity. LfgB was produced using 1 mM IPTG for 1 h and assayed with 15 mM $H_2O_2$. For the effect of RpoS (last two bars), cells were contacted with 20 mM $H_2O_2$ for 10 min to induce a stress response, prior to assaying for catalase activity. (B) MqsA purification via His tag (I) and EMSA results (II) showing that DNA-binding regulator MqsA binds the *lfg* operon. Mutating the *lfgA* promoter to interrupt the MqsA palindrome 5′-ACCG (N5) CGGT did not affect MqsA binding (left and middle, mutant probe 1), but mutating the region identified by the DNA-footprinting assay (bold red in Fig. S4) 252 bp upstream of the start codon, with nearby putative palindromic sequence 5′-ACAT (N2) ACAT (green highlight in Fig. S4), abolishes MqsA binding (right). (C) Two views of the predicted LfgB structure (UniProtKB: P52140). (D) SDS-PAGE demonstrating protease activity of LfgB toward MqsA. Purified proteins were mixed in enzyme reaction buffer and incubated overnight at 37°C. "M" indicates ladder.

Cryptic prophage CP4-57 has been linked to *E. coli* cell growth, biofilm formation, motility, and carbohydrate metabolism (48), and we previously found that the *lfgB* and *lfgA* deletions increase biofilm formation sixfold and twofold, respectively (48). In

addition, we found that the *lfgD* mutation reduces MqsR toxicity (40). Therefore, by characterizing protease LfgB, our results provide additional proof that cryptic prophages are beneficial and are involved in stress response (49). Note that the host has a tenuous relationship with its cryptic prophages since they increase environmental fitness (48, 49), including providing protection from acid stress through cryptic prophage CP4-57 (54-fold), as well as help cells resuscitate from the persister state by monitoring phosphate concentrations through CP4-57 regulator AlpA (50), and their lysis capabilities have to be silenced through CRISPR-Cas (51).

Our results also add another facet to MqsA regulation by finding a new protease that degrades MqsA. As previously demonstrated, Lon protease can degrade MqsA as well as other antitoxins under oxidative stress (12). In addition, ClpXP degrades MqsA in the absence of zinc that is used to stabilize the structure of MqsA, that is, when it is unfolded (52). It was proposed that the ClpX recognition site is accessible under non-stress conditions; however, under oxidative conditions, cysteine residues are oxidized, preventing the correct folding and the binding of zinc and allowing ClpXP to degrade MqsA (52). Hence, our results with protease LfgB provide additional evidence for the selective degradation of free antitoxins under stress conditions (12, 29, 52).

Our proposed mechanism is shown in Fig. 2. In the absence of stress, one physiological role of MqsA is to inhibit *rpoS* transcription (12), which is important for rapid growth. However, under stress conditions ($H_2O_2$, acid, and heat), Lon protease (12), ClpXP protease (52), and LfgB protease degrade antitoxin MqsA, facilitating the formation of RpoS and activation of the stress response. This also shifts the balance to MqsR (9, 12), which then performs differential mRNA decay (53), based on the presence of single-stranded, 5′-GCU sites (44). One example of differential mRNA decay is the degradation of the transcript for antitoxin GhoS, which results in activation of toxin GhoT (whose transcript lacks 5′-GCU sites) (14); this then allows toxin GhoT to reduce ATP and growth (54).

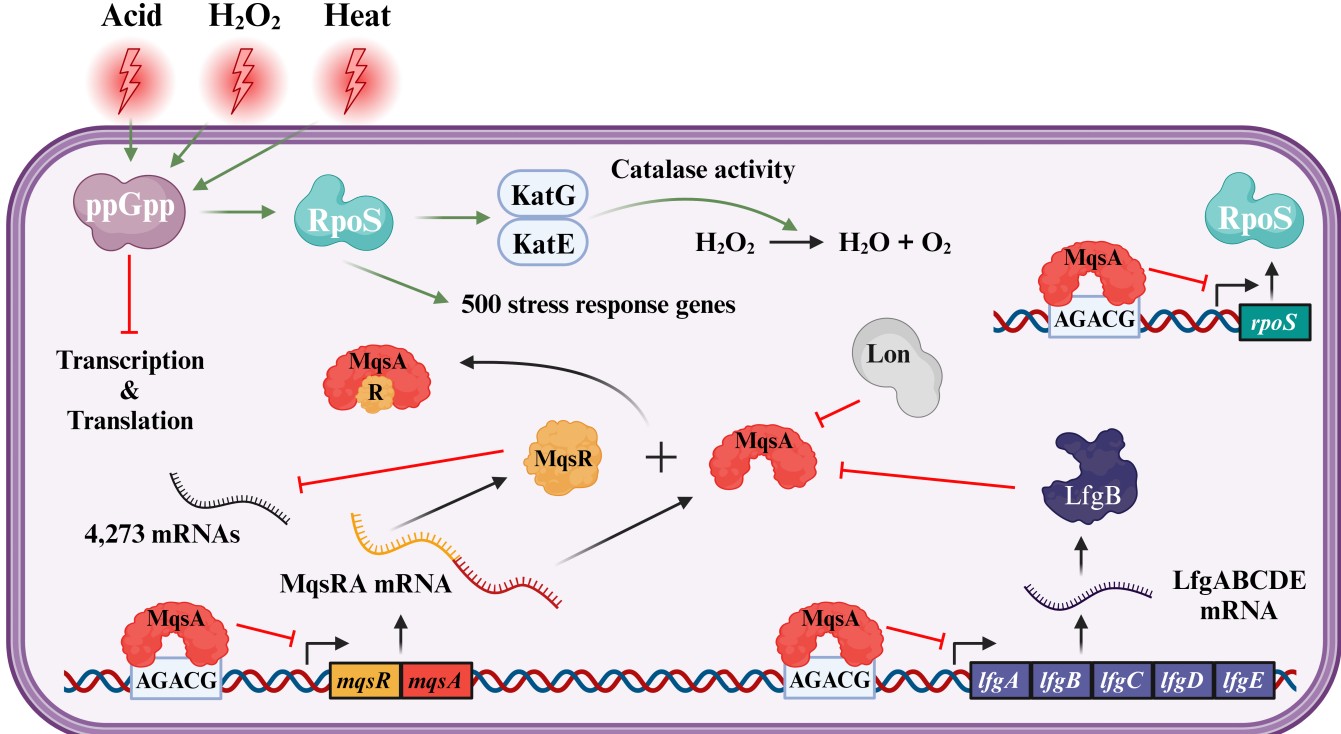

**FIG 2** Scheme for the MqsR/MqsA TA/LfgB protease stress response mechanism and its relationship with MqsA. Green arrows indicate activation, and red lines indicate inhibition.

Therefore, the type II TA system MqsR/MqsA is a multi-faceted regulator that facilitates growth of *E. coli* populations residing in the gut during exposure to bile (oxidative) stress. Since bile plays an important role as an interkingdom signal in the GI tract (55), our results also illustrate how a TA system can play an important role in host-microbe interactions by ensuring the survival of a commensal bacterium.

## MATERIALS AND METHODS

### Bacterial strains and growth conditions

The *E. coli* K-12 strains and plasmids used in this study are listed in Table 3. All cultures were grown in LB medium (56) at 37°C with 30 µg/mL of chloramphenicol to maintain the pCA24N plasmids.

### Single-cell transcriptome analysis

BW25113 and its unmarked isogenic mutant Δ*mqsRA* were harvested during exponential growth (turbidity of 0.8 at 600 nm), treated with 20 mM $H_2O_2$ for 10 min, and fixed with formaldehyde (1%) for 30 min. After centrifugation, cell pellets were washed with phosphate-buffered saline (PBS) and resuspended in 4:1 vol% methanol:glacial acetic acid and analyzed at the single-cell level as described previously (38).

### Viability assays with hydrogen peroxide, acid, and heat

Cells were cultured in LB to a turbidity of 0.8 at 600 nm and then exposed to 20 mM $H_2O_2$ for 10 min, acid conditions (pH 4) for 10 min, or heat (50°C) for 30 min. For cyclic exposure to acid (pH 2.5), cells were exposed four times for 10 min/cycle with 1-h growth in between each treatment.

**TABLE 3** Bacterial strains and plasmids used in this study[a]

| Strain | Genotype | Source |
|---|---|---|
| BW25113 | *rrnB3 ΔlacZ4787 hsdR514 Δ(araBAD)567 Δ(rhaBAD)568 rph-1* | (57) |
| BW25113 Δ*mqsRA* Δ*kan* | BW25113 Δ*mqsRA* ΔKm$^R$ | (53) |
| BW25113 Δ*lfgA* | BW25113 Δ*lfgA* Ω Km$^R$ | (57) |
| BW25113 Δ*gatR* | BW25113 Δ*gatR* Ω Km$^R$ | (57) |
| BW25113 Δ*yneL* | BW25113 Δ*yneL* Ω Km$^R$ | (57) |
| BW25113 Δ*ynfP* | BW25113 Δ*ynfP* Ω Km$^R$ | (57) |
| BW25113 Δ*yagA* | BW25113 Δ*yagA* Ω Km$^R$ | (57) |
| BW25113 Δ*holE* | BW25113 Δ*holE* Ω Km$^R$ | (57) |
| BW25113 Δ*yoeA* | BW25113 Δ*yoeA* Ω Km$^R$ | (57) |
| BW25113 Δ*yidL* | BW25113 Δ*yidL* Ω Km$^R$ | (57) |
| BW25113 Δ*ibpA* | BW25113 Δ*ibpA* Ω Km$^R$ | (57) |
| BW25113 Δ*ibpB* | BW25113 Δ*ibpB* Ω Km$^R$ | (57) |
| BW25113 Δ*lfgB* | BW25113 Δ*lfgB* Ω Km$^R$ | (57) |
| BW25113 Δ*ypjJ* | BW25113 Δ*ypjJ* Ω Km$^R$ | (57) |
| **Plasmid** | | |
| pCA24N | Cm$^R$; *lacI$^q$*, pCA24N | (58) |
| pCA24N-*lfgA* | Cm$^R$; *lacI$^q$*, pCA24N P$_{T5-lac}$::*lfgA* | (58) |
| pCA24N-*lfgB* | Cm$^R$; *lacI$^q$*, pCA24N P$_{T5-lac}$::*lfgB* | (58) |
| pCA24N-*mqsA* | Cm$^R$; *lacI$^q$*, pCA24N P$_{T5-lac}$::*mqsA* | (58) |
| pET28b | Km$^R$, expression vector with T7 promoter | Novagen |
| pET28b-*mqsA* | Km$^R$, lacI$^q$, pET28b P$_{T7-lac}$:: *mqsA* with *mqsA* C-terminus His-tagged | This study |

[a]Km$^R$ indicates kanamycin resistance, and Cm$^R$ indicates chloramphenicol resistance.

## Persister cell formation

Overnight cultures were grown to a turbidity of 0.8 at 600 nm, and then cells were resuspended in LB-ampicillin (100 µg/mL, 10 MIC) and incubated for 3 hour. Cells were washed twice with PBS, and viable cells were quantified using serial dilution and spot plating onto LB agar plates. Experiments were performed with at least three independent cultures (59).

## RNA structure prediction and DNA palindrome search

The RNA-predicted structures and palindrome search were obtained using the NCBI *E. coli* BW25113 genome sequence (NZ_CP009273.1), and the MFE RNA structures were predicted by the RNAfold webserver (http://rna.tbi.univie.ac.at/cgi-bin/RNAWeb-Suite/RNAfold.cgi).

## Quantitative real-time reverse-transcription PCR

Overnight cultures of Δ*mqsRA*/pCA24N and Δ*mqsRA*/pCA24N-*mqsA* were grown to a turbidity of 0.1 at 600 nm in LB/chloramphenicol medium, and then 1 mM of isopropyl β-D-thiogalactopyranoside (IPTG) for 30 min was used to induce expression of *mqsA*. In addition, overnight cultures of Δ*lfgB*/pCA24N and Δ*lfgB*/pCA24N-*lfgB* were grown to a turbidity of 0.5, and then 1 mM of IPTG was added for 1 hour to induce expression of *lfgB* to see the impact on *rpoS*. Then, cultures were incubated for 10 min with 20 mM hydrogen peroxide. Also, BW25113 and Δ*mqsRA* Δ*kan* were grown to a turbidity of 0.5, and then 20 mM $H_2O_2$ was added for 10 min. Cells were rapidly cooled in ethanol/dry ice and then centrifuged, and the pellets were collected with RNALater Buffer (Applied Biosystems, Foster City, CA, USA) to stabilize RNA. RNA was purified using the RNA Purification Kit (Roche). Quantitative real-time reverse transcription-PCR (qRT-PCR) was performed following the manufacturer's instructions for the iTaq Universal SYBR Green One-Step Kit (Bio-Rad) using 100 ng of total RNA as template. Primers were annealed at 60°C, and data were normalized against the housekeeping gene *rrsG* (13). The specificity of the qRT-PCR primers (Table S3) was verified via standard PCR, and fold changes were calculated using the method of Pfaffl (60) using the $2^{-\Delta\Delta CT}$.

## Proteolytic assay

Purified Lon, α-casein, MqsA, and LfgB were mixed in an enzyme reaction buffer (40 mM HEPES-KOH, 25 mM Tris-HCl, 4% sucrose, 4 mM dithiothreitol, 11 mM magnesium acetate, and 4 mM ATP) and incubated at 37°C for 3 h for α-casein degradation and overnight for MqsA degradation. SDS-PAGE was conducted using 5% stacking and 18% acrylamide resolving sections and staining following the manufacturer's instructions (Pierce Silver Stain Kit, Thermo Scientific).

## MqsA purification and EMSA

The *mqsA* coding region was amplified with primer pair pET28b-*mqsA*-F/R using MG1655 genomic DNA as the template. The amplified DNA fragment was purified, quantified, and ligated into pET28b digested with NcoI/HindIII. pET28b-*mqsA* was used to purify MqsA using standard methods (61). For DNA probes to investigate MqsA binding, the promoter region of *yfjY* was amplified with primer pair *yfjY*-P-F and *yfjY*-P-R, and the two mutant probes were also amplified with primer pairs *yfjY*-MP-F/*yfjY*-P-F and *yfjY*-MP2-F/ *yfjY*-P-F (Table S3). The probes were purified and labeled with biotin by using the Biotin 30-End DNA Labeling Kit (Thermo Scientific, Rockford, USA), and 0.25 pmol was used to assay the binding reaction with a series of concentrations of MqsA (62). The stopped reaction mixtures were run on a 6% polyacrylamide gel in Tris-borate EDTA and were then transferred to nylon membranes. The Chemiluminescence Nucleic Acid Detection Module Kit (Thermo Scientific) was used to observe the shift of the DNA probes on the membranes.

## DNase I footprinting assay

This assay was conducted as reported previously (62). The FAM-labeled probe covering the promoter region of *yfjY* was amplified with primer pair FAM-*yfjY*-P-F and *yfjY*-P-R, and the products were purified with QIAEX II Gel Extraction Kit (Qiagen, Hilden, Germany). The labeled probes (200 ng) were mixed with varying amounts of MqsA, and the mixtures were incubated for 30 min at 25°C. An orthogonal combination of DNase I (NEB, M0303S) and incubation time were used to achieve the best cutting efficiency. A final concentration of 200 mM EDTA was added to the reaction mixture to stop the reaction. The DNA was purified again with a QIAEX II Gel Extraction Kit (Qiagen, Hilden, Germany), and the generated products were screened and analyzed as reported (62).

## Catalase assay

LfgB was produced in exponentially growing cells (turbidity of 0.1 at 600 nm) using 1 mM IPTG for 1 hour. For the effect of RpoS, cells were contacted with 20 mM $H_2O_2$ prior to performing the catalase assay to induce a stress response. Catalase activity was determined spectrophotometrically by recording the decrease in the absorbance of $H_2O_2$ at 240 nm in a UV/visible spectrophotometer as described previously (63). Briefly, five independent cultures per strain were grown overnight, 1 mL aliquots were taken, and cells were collected by centrifugation for 1 min at 13,000 rpm, washed with sterile HEPES buffer (50 mM, pH 7.5), centrifuged again, frozen with liquid nitrogen, and stored at −70°C. Thawed pellets were resuspended in 1 mL of sterile cold HEPES buffer (50 mM, pH 7.5) with $MgCl_2$ 10 mM and 0.025% Triton X-100 and disrupted by sonication using two pulses of 20 sec with 1-min pause between cycles. Catalase activity was determined using 15 mM $H_2O_2$ as substrate and normalized based on the protein level in the cell extracts as determined using the Bradford method.

## Mass spectroscopy

Mass spectroscopy was used to identify the sequences of purified LfgB as previously described (64). In brief, the putative LfgB bands were excised from the SDS-PAGE gel and digested with trypsin. The peptide fragments were then analyzed with liquid chromatography tandem mass spectrometry, and the data were compared against the LfgB sequence.

## ACKNOWLEDGMENTS

We are grateful for the Keio and ASKA strains provided by the National Institute of Genetics of Japan and for the single-cell transcriptomic experiment by International Flavors and Fragrances. We also appreciate the help of Erin Essington with the figures.

This work was supported by a Fulbright Scholar Fellowship and GAIN (GAIN, Xunta de Galicia) grant IN606A-2020/035 for L.F.-G. and a National Research Foundation of Korea (NRF) grant from the Korean Government (NRF-2020R1F1A1072397) for S.S. This study also was funded by the grants PI19/00878 and PI22/00323 awarded to M. Tomás within the State Plan for R+D+I 2013-2016 (National Plan for Scientific Research, Technological Development and Innovation 2008–2011) and co-financed by the ISCIII-Deputy General Directorate for Evaluation and Promotion of Research-European Regional Development Fund "A way of Making Europe" and Instituto de Salud Carlos III FEDER.

## AUTHOR AFFILIATIONS

[1]Department of Chemical Engineering, Pennsylvania State University, University Park, Pennsylvania, USA

[2]Microbiology Department, Hospital A Coruña (HUAC), A Coruña, Spain

[3]Microbiology Translational and Multidisciplinary (MicroTM) - Research Institute Biomedical A Coruña (INIBIC) and Microbiology, University of A Coruña (UDC), A Coruña, Spain

⁴Key Laboratory of Tropical Marine Bio-resources and Ecology, Institute of Oceanology, Chinese Academy of Sciences, Nansha, Guangzhou, China

⁵Guangdong Key Laboratory of Marine Materia Medica, Chinese Academy of Sciences, Nansha, Guangzhou, China

⁶Innovation Academy of South China Sea Ecology and Environmental Engineering, South China Sea, Chinese Academy of Sciences, China, Nansha,, Guangzhou, China

⁷University of Chinese Academy of Sciences, Beijing, China

⁸Department of Animal Science, Jeonbuk National University, Jeonju-Si, Jellabuk-Do, South Korea

⁹Department of Agricultural Convergence Technology, Jeonbuk National University, Jeonju-Si, Jellabuk-Do, South Korea

¹⁰Departamento de Microbiología y Parasitología, Facultad de Medicina, Universidad Nacional Autónoma de México, Mexico, Mexico

¹¹Southern Marine Science and Engineering Guangdong Laboratory (Guangzhou), Nansha, Guangzhou, China

## AUTHOR ORCIDs

Laura Fernández-García  http://orcid.org/0000-0002-8531-6105
Rodolfo Garcia-Contreras  http://orcid.org/0000-0001-8475-2282
Maria Tomas  http://orcid.org/0000-0003-4501-0387
Yunxue Guo  http://orcid.org/0000-0002-2006-1746
Xiaoxue Wang  http://orcid.org/0000-0002-7257-1916
Thomas K. Wood  http://orcid.org/0000-0002-6258-529X

## FUNDING

| Funder | Grant(s) | Author(s) |
| --- | --- | --- |
| FULBRIGHT \| Fulbright US Scholar Program | | Laura Fernández-García |
| Gain Xunta de Galicia | IN606A-2020/035 | Laura Fernández-García |
| National Plan for Scientific Research | PI19/00878 and PI22/00323 | Maria Tomas |

## AUTHOR CONTRIBUTIONS

Laura Fernández-García, Conceptualization, Data curation, Formal analysis, Funding acquisition, Investigation, Methodology, Writing – review and editing | Xinyu Gao, Investigation, Methodology | Joy Kirigo, Investigation | Sooyeon Song, Investigation | Michael E. Battisti, Investigation | Rodolfo Garcia-Contreras, Investigation, Writing – review and editing | Yunxue Guo, Investigation, Methodology, Project administration, Resources, Supervision, Validation, Writing – review and editing | Xiaoxue Wang, Resources, Supervision, Writing – review and editing | Thomas K. Wood, Conceptualization, Formal analysis, Funding acquisition, Methodology, Project administration, Resources, Supervision, Validation, Writing – original draft, Writing – review and editing.

## DATA AVAILABILITY

The data underlying this article are available in the article and in its supplemental material.

## ADDITIONAL FILES

The following material is available online.

## Supplemental Material

**Supplemental material (Spectrum03471-23 final SI.docx).** Tables S1 to S3; Figures S1 to S7.

## Open Peer Review

**PEER REVIEW HISTORY (review-history.pdf).** An accounting of the reviewer comments and feedback.

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
