## [Reviewer comments · Microbiology Spectrum]

Microbiology Spectrum

Single-Cell Analysis Reveals Cryptic Prophage Protease LfgB Protects *Escherichia coli* During Oxidative Stress by Cleaving Antitoxin MqsA

Laura Fernández-García, Xinyu Gao, Joy Kirigo, Sooyeon Song, Michael Battisti, Rodolfo Garcia-Contreras, Maria Tomas, Yunxue Guo, Xiaoxue Wang, and Thomas Wood

Corresponding Author(s): Thomas Wood, The Pennsylvania State University

Review Timeline:

Submission Date:	September 23, 2023
Editorial Decision:	October 16, 2023
Revision Received:	November 29, 2023
Accepted:	November 30, 2023

Editor: Ilana Kolodkin-Gal

Reviewer(s): Disclosure of reviewer identity is with reference to reviewer comments included in decision letter(s). The following individuals involved in review of your submission have agreed to reveal their identity: Robert JC McLean (Reviewer #1)

Transaction Report:

DOI: <https://doi.org/10.1128/spectrum.03471-23>

October 16, 2023

Prof. Thomas K. Wood
The Pennsylvania State University
Chemical Engineering
304 Chemical and Biochemical Engineering
UNIVERSITY PARK, Pennsylvania 16802

Re: Spectrum03471-23 (Single-Cell Analysis Reveals Cryptic Prophage Protease LfgB Protects Escherichia coli During Oxidative Stress by Cleaving Antitoxin MqsA)

Dear Prof. Thomas K. Wood:

Dear Prof. Wood,

During the revision, please carefully address utilizing the appropriate experiments the major concerns of both reviewers, especially the concerns raised by reviewer 2

Link Not Available

Sincerely,

Ilana Kolodkin-Gal

Journals Department
Reviewer comments:

Reviewer #1 (Comments for the Author):

This is a very interesting and timely study in which the authors have identified a new role for a toxin-antitoxin system during E. coli growth and its influence in regulating oxidative stress. There are a number of minor questions that I would like the authors to address:

- 1) Have the authors done any investigations in an rpoS negative background? This data would greatly strengthen the experimental interpretation.
- 2) The authors encountered some issues with the low solubility of the LfgB protein. Might one possible explanation be that this protein is membrane-associated?
- 3) In lines 144-145, the authors make the claim that cryptic prophages are beneficial and involved in the stress response. I would strongly recommend softening this claim since the authors have investigated a subset of prophages and not all prophages.
- 4) As a minor point, please insert "(TA)" without the quotation marks following "Toxin/antitoxin" in the opening sentence of the introduction. That will nicely define this acronym.

Reviewer #2 (Comments for the Author):

The manuscript by Fernández-García describes the possible role of the MqsRA toxin-antitoxin system and the LfgB protease of CP4-57 cryptic prophage in response to oxidative stress in *E. coli*. They first show that the mqsRA deletion mutant is more sensitive to oxidative stress than the wild type strain, and identified several genes that are induced in the wild type strain when compared to the mqsRA mutant following H₂O₂ exposure. Among these, genes of the lfgABCDE operon of the cryptic prophage CP4-57 are the focus of this study. They authors show that both lfgA and lfgB mutations prevented cells from surviving oxidative stress, and that the lfgB mutant had a reduced catalase activity and was more sensitive heat and acid treatments. The link between MqsRA and the putative LfgB protease was further investigated, showing that MqsA can bind and repress transcription of the lfg operon. Based on the possible degradation of MqsA by LfgB observed in vitro, the authors conclude that LfgB controls the oxidative stress response through MqsA degradation.

Although the model is interesting, it is not supported by the data presented.

Comments:

- The main conclusion that MqsA is a substrate of the putative LfgB protease in vitro is not supported by the data and there is no evidence showing that MqsA is accumulating in the absence of lfgB in vivo. The important in vitro experiments from Figure 2 lack all the controls and does not show proper kinetics of degradation under reasonable experimental conditions. How can the authors be confident that LfgB, but not any other proteases from the actual purification, degrades MqsA? How is an overnight incubation at 37°C relevant in this case? How were the MqsA degradation products identified? Such critical issues should have been properly addressed.
- Complementation of the H₂O₂ sensitivity of the mqsRA mutant was not performed.
- There is no evidence that the lfgB transcript is cleaved by MqsR.

Staff Comments:

Preparing Revision Guidelines

Please return the manuscript within 60 days; if you cannot complete the modification within this time period, please contact me. If you do not wish to modify the manuscript and prefer to submit it to another journal, please notify me of your decision immediately so that the manuscript may be formally withdrawn from consideration by Microbiology Spectrum.

If your manuscript is accepted for publication, you will be contacted separately about payment when the proofs are issued;

please follow the instructions in that e-mail. Arrangements for payment must be made before your article is published. For a complete list of **Publication Fees**, including supplemental material costs, please visit our website.

Spectrum03471-23: Single-Cell Analysis Reveals Cryptic Prophage Protease LfgB Protects Escherichia coli During Oxidative Stress by Cleaving Antitoxin MqsA

We wish to thank the editor and two reviewers for their help with this manuscript and feel the manuscript is improved due to the careful review. We have addressed all the issues as indicated below (our comments are underlined and the changed text is highlighted yellow in the revised manuscript). Note the line numbers below refer to the *revised* text.

Specifically, we have conducted 5 additional experiments to (i) determine the impact of H₂O₂ and MqsRA on *yjx* transcription via qRT-PCR for BW25113 vs. BW25113 *mqsRA* ΔKan using population averages rather than single cells and found the changes in *yjx* transcription can only be detected via the single-cell approach (results added to line 90 and raw data in **Table S2A**), (ii) determine the impact of LfgB production on *rpoS*⁺ transcription via qRT-PCR, which shows 3X induction *rpoS*⁺ when LfgB is produced due to likely degradation of MqsA and the derepression of *rpoS*⁺ (**Table S2C**, line 148), (iii) determine the impact of LfgB on catalase activity in the absence of RpoS and found LfgB, as expected, has less effect in the absence of RpoS (**Fig 1** and line 151), (iv) add purified Lon and alpha-caesin/BSA as positive controls for the protein gel (**Fig. S6**), and (v) use mass spectrometry to show the confirm LfgB was purified correctly (**Fig. S7**).

We also improved the main figures by consolidating the results and by adding the catalase results to **Fig. 1** and re-drew the mechanism schematic of **Fig. 2** to increase clarity.

Reviewer 1

This is a very interesting and timely study in which the authors have identified a new role for a toxin-antitoxin system during *E. coli* growth and its influence in regulating oxidative stress. There are a number of minor questions that I would like the authors to address:

1) Have the authors done any investigations in an *rpoS* negative background? This data would greatly strengthen the experimental interpretation.

As suggested, we measured investigated the effect of LfgB on catalase activity in the absence of RpoS by producing LfgB via BW25113 *rpoS*/pCA24N-lfgB vs. BW25113/pCA24N-LfgB and found, as expected, that in the absence of RpoS, LfgB is less effective in increasing catalase. This is most likely due to the fact that there is no activation of RpoS by LfgB degrading MqsA, which represses *rpoS* (**Fig. 1A**, line 151)

2) The authors encountered some issues with the low solubility of the LfgB protein. Might one possible explanation be that this protein is membrane-associated?

As suggested, we have added your idea to the manuscript (line 146)

3) In lines 144-145, the authors make the claim that cryptic prophages are beneficial and involved in the stress response. I would strongly recommend softening this claim since the authors have investigated a subset of prophages and not all prophages.

As suggested, to strengthen this claim, we now refer to our three additional, earlier studies in which we (i) deleted all 9 *E. coli* cryptic prophages (166 kb, so we have studied all 9 *E. coli* cryptic prophages) to show cryptic prophages increase environmental fitness (Nature Commun 2010, cited over 600 times) and also refer to results from this 2010 manuscript that show the same cryptic prophage in this current paper that encodes LfgB, CP4-57, increases fitness for acid stress by 54-fold⁴, (ii) found that, although cryptic prophages increase environmental fitness, their residual lytic capabilities must be silenced by CRISPR-Cas², and (iii) discovered CP4-57 helps the cell resuscitate from the persister state by monitoring phosphate concentrations through regulator AlpA¹. (line 165).

4) As a minor point, please insert "(TA)" without the quotation marks following "Toxin/antitoxin" in the opening sentence of the introduction. That will nicely define this acronym.

As suggested, we have defined the acronym here (line 25), and it was previously defined in the Abstract (line 2).

Reviewer 2

The manuscript by Fernández-García describes the possible role of the MqsRA toxin-antitoxin system and the LfgB protease of CP4-57 cryptic prophage in response to oxidative stress in *E. coli*. They first show that the *mqsRA* deletion mutant is more sensitive to oxidative stress than the wild type strain, and identified several genes that are induced in the wild type strain when compared to the *mqsRA* mutant following H₂O₂ exposure. Among these, genes of the *lfgABCDE* operon of the cryptic prophage CP4-57 are the focus of this study. They authors show that both *lfgA* and *lfgB* mutations prevented cells from surviving oxidative stress, and that the *lfgB* mutant had a reduced catalase activity and was more sensitive heat and acid treatments. The link between MqsRA and the putative LfgB protease was further investigated, showing that MqsA can bind and repress transcription of the *lfg* operon. Based on the possible degradation of MqsA by LfgB observed in vitro, the authors conclude that LfgB controls the oxidative stress response through MqsA degradation.

Although the model is interesting, it is not supported by the data presented.

The reviewer is, of course, correct, and so we strive now to indicate more clearly which parts of the proposed mechanism are speculative (e.g., lines 135 by adding “likely” and line 141, “...we cannot strictly rule out...”). However, this is the first use of single-cell transcriptomics for the TA field and one of the first single-cell transcriptomic studies that actually provides some new biological insight (all reports to date have just confirmed existing, whole population physiology). Moreover, we have conducted new experiments that corroborate our results by showing, via the new PCR study, that we cannot detect changes in *lfgB* in population-averaged cells, and so, single-cell transcriptomics are necessary for link MqsRA to previously-uncharacterized *lfgB* (line 90).

Therefore, herein, we (i) derived from single-cell analysis one of the first, if not the first, physiologically-relevant new insights for this nascent method for procaryotes, (ii) applied single-cell analysis to TA systems for the first time, (iii) studied the wild-type and deletion mutant so no TA overexpression is involved, (iv) discovered and characterized an operon in a cryptic prophage that is used by the host in its oxidative stress response, and (v) found a physiological role for the type II TAs, MqsR/MqsA (H₂O₂ stress response) when most studies fail to find a physiological role (other than phage inhibition, which we discovered first in 1996), and prominent labs (e.g. Laub and Van Melderen) incorrectly have concluded there is not expression for these ubiquitous TAs (except the Laub lab has shown the Van Melderen lab was incorrect, at least in the incorrect lack of promoter activity for *mqsRA* and the Laub lab has contradicted itself by finding the physiological role of MqsRAC for phage inhibition where it claimed no protein was produced previously with this TAs). Hence, our results will have a dramatic impact on the TA field by demonstrating clearly a role in cell physiology distinct from phage inhibition as well as revealing more about how the tools of the former enemy of the cell, cryptic prophages, are incorporated into the host stress response.

So although we have not fully elucidated the mechanism, we think that our accomplishments are considerable for a single paper.

Comments:

1. The main conclusion that MqsA is a substrate of the putative LfgB protease in vitro is not supported by the data and there is no evidence showing that MqsA is accumulating in the absence of LfgB in vivo.

a. As suggested, we added ‘likely’ through MqsA degradation (line 135) since our protease work is imperfect due to poor LfgB activity (although we have tried repeatedly and with protein tags); however, we do demonstrate some degradation of MqsA in the presence of purified LfgB in Fig. 2 and have now added both mass spectroscopy data to confirm LfgB was purified and SDS-PAGE controls as indicated below.

b. The suggested experiment of quantifying MqsA accumulation in single cells is nearly impossible, i.e., showing MqsA accumulation in a *lfgB* mutant, as that would require technology to see MqsA levels in single cells and to our knowledge, single cell proteomics is not that developed yet. Moreover, we also feel this is unnecessary since the whole point of this manuscript is to discover new biology from gene expression in single cells, and we did that successfully (probably for the first time in any lab and certainly the first time for toxin/antitoxin systems).

The important in vitro experiments from Figure 2 lack all the controls and does not show proper kinetics of degradation under reasonable experimental conditions.

c. As suggested, along with showing previously MqsA in the absence of protease LfgB, we now show LfgB has protease activity on α -casein and BSA as well as use purified Lon protease as positive control with α -casein and BSA (Fig. S6). We recognize these protein gels are rough, but degradation can be seen by seeing the reduction of the α -casein and BSA bands with both Lon and LfgB.

2. How can the authors be confident that LfgB, but not any other proteases from the actual purification, degrades MqsA?

The reviewer is correct but the experiments required for this control are beyond the scope of this manuscript. Therefore, we added to the manuscript “although we cannot strictly rule out that other proteases are present in the purified LfgB purification.” (line 141).

In addition, we now show via mass spectrometry data that LfgB was purified correctly as it clearly shows we purified LfgB well and clearly shows its degradation, likely due to self-digestion (Fig. S7). These data rule out nearly completely that background proteases are responsible for the degradation of MqsA seen in Fig. 1D. (line 144).

3. How is an overnight incubation at 37°C relevant in this case?

As suggested, we now indicate now indicate the LfgB may be membrane-associated and that is why we do not see robust protease activity. (line 146)

4. How were the MqsA degradation products identified? Such critical issues should have been properly addressed.

Unfortunately, we were unable to assay MqsA degradation products; however, we were able to use mass spectrometry to determine how protease LfgB degrades itself and have added these results to Fig. S7.

5. Complementation of the H₂O₂ sensitivity of the mqsRA mutant was not performed.

We note that complementation of the *lfgB* mutation for the H₂O₂ phenotype was made (see Fig. S2) since LfgB is the primary focus of this manuscript. Complementation of the *mqsRA* mutant H₂O₂ and acid phenotypes along with data showing production of MqsA reduces peroxidase activity is not necessary here as we previously published this in Fig. 2bcd of Wang et al. 2011³, and we now indicate that in the text (line 76).

6. There is no evidence that the *lfgB* transcript is cleaved by MqsR.

We agree and that is why we already qualified our statement as “MqsR may degrade the mRNA containing *lfgB*” (line 122), and we also do not show MqsR cleaving the *lfgB* mRNA in the schematic (Fig. 2). We merely tried to indicate where MqsR could possibly cleave *lfgB* mRNA by identifying likely single-stranded regions (Fig. S3).

REFERENCES

1. **Song S, Kim J-S, Yamasaki R, Oh S, Benedik MJ, et al.** 2021. *Escherichia coli* cryptic prophages sense nutrients to influence persister cell resuscitation. *Environ Microbiol* **23**:7245-7254.
2. **Song S, Semenova E, Severinov K, Fernández-García L, Benedik MJ, et al.** 2022. CRISPR-Cas Controls Cryptic Prophages. *International Journal of Molecular Sciences* **23**:16195.
3. **Wang X, Kim Y, Hong SH, Ma Q, Brown BL, et al.** 2011. Antitoxin MqsA helps mediate the bacterial general stress response. *Nat Chem Biol* **7**:359-366.
4. **Wang X, Kim Y, Ma Q, Hong SH, Pokusaeva K, et al.** 2010. Cryptic prophages help bacteria cope with adverse environments. *Nat Commun* **1**:147.

Re: Spectrum03471-23R1 (Single-Cell Analysis Reveals Cryptic Prophage Protease LfgB Protects Escherichia coli During Oxidative Stress by Cleaving Antitoxin MqsA)

Dear Prof. Thomas K. Wood:

Many thanks for addressing the comments of both reviewers to their full satisfaction. It was a pleasure reading the revised manuscript.

Your manuscript has been accepted, and I am forwarding it to the ASM production staff for publication. Your paper will first be checked to make sure all elements meet the technical requirements. ASM staff will contact you if anything needs to be revised before copyediting and production can begin. Otherwise, you will be notified when your proofs are ready to be viewed.

Sincerely,
Ilana Kolodkin-Gal
Editor
Microbiology Spectrum